# Challenges and Opportunities in Aligning Conservation with Development in China’s National Parks: A Narrative Literature Review

**DOI:** 10.3390/ijerph191912778

**Published:** 2022-10-06

**Authors:** Andrew Rule, Sarah-Eve Dill, Gordy Sun, Aidan Chen, Senan Khawaja, Ingrid Li, Vincent Zhang, Scott Rozelle

**Affiliations:** Stanford Center on China’s Economy and Institutions, Stanford University, Stanford, CA 94305, USA

**Keywords:** protected area, national park, biodiversity, social-ecological systems, China

## Abstract

As part of its effort to balance economic development with environmental objectives, China has established a new national park system, with the first five locations formally established in 2021. However, as the new parks all host or are proximate to human populations, aligning the socioeconomic needs and aspirations of local communities with conservation aims is critical for the long-term success of the parks. In this narrative review, the authors identify the ecological priorities and socioeconomic stakeholders of each of the five national parks; explore the tensions and synergies between these priorities and stakeholders; and synthesize the policy recommendations most frequently cited in the literature. A total of 119 studies were reviewed. Aligning traditional livelihoods with conservation, limiting road construction, promoting education and environmental awareness, and supporting the development of a sustainable tourism industry are identified as important steps to balance conservation with economic development in the new national parks.

## 1. Introduction

The world is undergoing rapid biodiversity and ecosystem loss, driven by large-scale anthropogenic impacts such as landscape degradation, deforestation, and pollution. Some scholars have termed this ongoing ecological crisis the “Sixth Mass Extinction” [1]. The last such catastrophe occurred over 65 million years ago and led to the disappearance of the dinosaurs and many other species. Addressing this escalating crisis is integral to achieving sustainable development [2]. Among other initiatives, working toward sustainability requires the protection and restoration of large tracts of terrestrial and marine ecosystems. Indeed, these needs have been explicitly codified in the United Nations Sustainable Development Goals (SDGs). For instance, SDG 15 aims to: “protect, restore and promote sustainable use of terrestrial ecosystems, sustainably manage forests, combat desertification, and halt and reverse land degradation and halt biodiversity loss” [3].

Many countries have responded through expansive actions, including China, the world’s most populous and economically largest (as measured in gross domestic product at purchasing power parity) country. To balance economic development with environmental objectives, China’s government has enacted numerous policies and allocated high levels of support into ecological protection and restoration in recent decades [4]. While this led to significant improvements in many key ecosystem services and other indicators [5], many problems remain, including continued biodiversity loss [6]. In response, China has begun to expand and consolidate its system of protected areas (PAs) to prevent habitat degradation and destruction [7]. For instance, as part of its commitment to the Convention on Biological Diversity, Beijing is set to establish official environmental protection over at least 30% of its land by 2030 [8]. China is one of 20 like-minded megadiverse countries (LMMCs) that together encompass nearly 29% of the planet’s terrestrial surface and host more than half of IUCN-listed threatened species [2]. Meeting this goal is therefore critical not only to the country’s own sustainability, but also for global ecological conservation generally.

A significant measure China has undertaken to meet its ambitious conservation goals is its new national park system with the first five locations formally established in 2021 [9]. While the country already has established a network of PAs over the past several decades [10], these have suffered from shortcomings of design and enforcement, leading to suboptimal ecological outcomes, a challenge common for conservation efforts worldwide [11]. China’s new national parks are meant to address these issues, institutionalizing a more rigorous approach [12]. However, as in other large developing countries, many of the most ecologically important, biodiverse areas host or are proximate to large human populations. Aligning the socioeconomic needs and aspirations of these often rural and traditional communities with conservation aims is critical for the long-term success of PAs [13]. How China’s first five national parks address these challenges are important not only for the parks themselves, but also for system expansion in the future and for PA management more broadly. 

To this end, it is necessary to view China’s national parks from a social-ecological systems (SESs) perspective in order to understand the challenges and potential solutions they face in aligning the imperatives of conservation with those of economic and social development. An SES perspective posits that human communities and organizations are inextricably embedded in nature, and that their actions both influence and are influenced by the ecological conditions around them [14]. This iterative process of feedbacks determines community wellbeing, ecological integrity, and the resilience and trajectory of the system as a whole. National park governance must take these dynamics into account.

There exists a small body of international literature that examines the relationship between conservation and development in PAs. As noted in Mathevet et al. [15], the success of ecological conservation and socioeconomic development projects in PAs are often so interdependent that building solidarity between local communities and park authorities should be a crucial priority for PA management. This solidarity might take the form of mutual trust between communities and park authorities, participatory- and dialogue-based management, or recognition of different knowledge systems and values in park governance. Other studies of PAs, including several in developing countries, have provided further evidence for the importance of these values. For instance, a survey of 114 African and European PA managers conducted by Gatiso et al. [16] found that a majority of the PAs played a positive role in reconciling conservation with socioeconomic development, although fewer synergies and more trade-offs were found in the African PAs. The researchers further found that synergies were more likely to arise in PAs where park authorities were more empowered and local communities were involved in management and planning. A study in Indonesia by Gurney et al. [17] focused on integrated PAs, or PAs that were specifically designed to alleviate poverty by prioritizing environmental education and improved access to drinking water alongside conservation goals. The PAs were shown to initially reduce poverty, but this effect diminished over time because of a lack of effective follow-through, highlighting both the benefits and risks of attempting to address interrelated socioeconomic and ecological needs in PA design.

However, little research of this kind has taken place in China, despite the existence of vast areas of protected land in the country where the imperatives of conservation and socioeconomic development are frequently at odds. While some studies have examined the development of specific protected areas in China through the lens of the socioeconomic needs of local communities [18,19,20,21], there is a notable lack of scholarship that considers this question from the larger perspective of China’s protected area system as a whole. Moreover, as China’s national park system has only existed for a brief time, no prior studies have focused on the ways in which the new system interacts with socioeconomic stakeholders within the national parks specifically, to the best of our knowledge. Therefore, there is a need for research that can serve to guide these interactions between park authorities and socioeconomic stakeholders as the new park system continues to expand.

The present article meets this need by reviewing the literature on the ecological and socioeconomic conditions of China’s first five national parks, focusing on the period leading up to their formal establishment and their first months of existence. In particular, we identify the key conservation objectives of each park, the relevant stakeholders, and the potential tensions and synergies between them. We also document and discuss the policies most frequently recommended in the literature as ways to manage the interactions between the environment of the parks and nearby local communities. Finally, we synthesize our findings and advance a set of policy suggestions to align economic and social development with conservation in the park system as a whole.

The following section provides a brief overview of each park’s ecological and socioeconomic background. In Section 3, we outline the methodology of our narrative review, the results of which we present in Section 4. In Section 5, we discuss our findings and offer our recommended set of future policy directions before concluding in Section 6.

## 2. Overview of China’s First Five National Parks

China’s national park system was first formally proposed in 2013, after which ten parks were successively rolled out on a pilot basis between 2016 and 2019 [7,22]. These parks were placed under the management of the newly formed National Park Administration, a subdivision of the National Forestry and Grasslands Administration, bringing China’s formerly fragmented and decentralized system of protected areas under a single governing body [10]. The first five parks to graduate from pilot status to official park status, Sanjiangyuan National Park, Wuyishan National Park, Hainan Tropical Rainforest National Park, Giant Panda National Park, and Northeast Tiger, and Leopard National Park, were formally established and opened to the public in the fall of 2021. The parks are operated by local administrations that are overseen jointly by the provincial governments and the system-wide National Park Administration [22], although the specific management structure varies for each park, as described below. In this section, we provide a brief overview of these first five national parks, including their geography, ecological characteristics, socioeconomic conditions, and management structures.

*Sanjiangyuan National Park (SNP):* Launched as a pilot site in 2016 and formally established as a national park in 2021, SNP is primarily managed by the provincial government of Qinghai Province through a local governing body, the Sanjiangyuan National Park Administration [22]. It is the largest protected area in the world, covering an area 14 times larger than Yellowstone National Park [23] (Figure 1). Located on the Qinghai–Tibetan Plateau, the park’s environment is comprised of alpine forests, grasslands, and meadows at an average elevation of more than 13,000 feet above sea level. Crucial for both national and continental hydrology, SNP also houses the headwaters of three of Asia’s major rivers: the Yangtze, the Yellow River, and the Lancang (Mekong River). Within its boundaries roam the world’s largest population of snow leopards (*Panthera uncia*), among other vulnerable and endangered species. SNP is also the site of over 200 major Tibetan Buddhist monasteries, many over a millennium old, and remains home to numerous pastoralist and farming communities. The counties intersecting with the park have some of the highest poverty rates in China and are remote from major population centers [24].

*Wuyishan National Park (WNP):* WNP was founded as a pilot site in 2017 and formally established as a national park in 2021. It is administered by a local body, the Wuyishan National Park Administration, under the oversight of the system-wide National Park Administration [22]. It straddles the provinces of Jiangxi and Fujian and is located in the highest mountain range in southeastern China (Figure 2). Rich in biodiversity, especially for plants, reptiles, and amphibians, the park is largely composed of rugged mountainous areas and forms an important part of the region’s hydrological system through its streams and rivers, especially the Min River, which empties into the East China Sea near Fuzhou [25]. WNP covers a culturally rich area, particularly known for its historical contributions to Confucianism, and is designated as an UNESCO World Heritage Site. The region is also famous for tea (among other crops), and tea cultivation remains an important part of the rural economy [26].

*Hainan Tropical Rainforest National Park (HTRNP):* Founded as a pilot site in 2019 and officially established as a national park in 2021, HTRNP is administered primarily by the provincial government of Hainan Province through the Hainan Tropical Rainforest National Park Administration [22]. It covers approximately 15% of the island province of Hainan (Figure 3). The park protects China’s largest expanse of tropical rainforest, much of which is virgin forest [27]. HTRNP covers mountainous terrain, including the Wuzhi Mountains, which has the island’s highest peak. The park is home to the Hainan gibbon (*Nomascus hainanus*), an endemic species and the world’s most endangered primate. The area surrounding the park has historically been rural and includes significant plantations for rubber, which is a major cash crop. Located inland, HTRNP is relatively remote from the tourist and commercial hubs of Haikou and Sanya.

*Giant Panda National Park (GPNP):* GPNP was founded as a pilot site in 2017 [28]. In 2021, it was officially established as a national park under the management of the Giant Panda National Park Administration, which is jointly overseen by the provincial and national governments [22]. The park is located discontinuously across three provinces, Sichuan, Gansu, and Shaanxi, and covers a total area three times the size of Yellowstone (Figure 4). Its boundaries correspond to the habitat range of the giant panda (*Ailuropoda melanoleuca*) [29]. The park covers diverse ecosystems but focuses in particular on bamboo-rich mountainous landscapes. GPNP is also a refuge for the red panda (*Ailurus fulgens*) and the golden snub-nosed monkey (*Rhinopithecus roxellana*), both endangered and endemic to China. While the park’s inhabitants are widely dispersed across many rural farming communities, its boundaries also abut the northern edges of the Chengdu metropolitan area. Of the five national parks, GPNP has the highest number of people living within its boundaries [29].

*Northeast Tiger and Leopard National Park (NTLNP):* Founded as a pilot site in 2016 and formally established in 2021, NTLNP is governed by the Northeast Tiger and Leopard National Park Administration under the direct oversight of the national government [22]. The park is located in Jilin and Heilongjiang provinces, at an international junction with Russia and North Korea [30] (Figure 5). It is approximately 1.6 times the size of Yellowstone. The park is built around the last remaining populations of the Amur tiger (*Panthera tigris tigris*) and Amur leopard (*Panthera pardus orientalis*), two of the most endangered felids in the world, whose ranges also extend into Russia. The two species anchor an ecoregion of conifers and broadleaf trees. The communities surrounding the national park are largely rural and have historically relied on farming and timber as economic mainstays.

## 3. Materials and Methods

We conducted a narrative review synthesizing past research into the relationship between conservation and development in each of the five national parks. For each park, we conducted an initial literature search using Scopus and ProQuest’s Agricultural and Environmental Science Collection (AESC). In this initial search, we limited our results to papers that (1) were initially published in English, (2) contained original, peer-reviewed empirical research, and (3) were published no earlier than 2012. A complete list of search terms used in the initial search can be found in Appendix A (Table A1).

We then strictly applied our inclusion criteria to generate a final list of articles for review. Two key criteria were used to identify relevant articles. First, we only considered research that took place inside or in the immediate vicinity of the five national parks, including research that took place in those areas before the parks were officially founded. Second, we only selected articles that focused primarily on the interface between conservation efforts and socioeconomic stakeholders within the parks. Our final list of articles for review contained 28 studies of Sanjiangyuan National Park, 12 studies of Wuyishan National Park, 19 studies of Hainan Tropical Rainforest National Park, 39 studies of Giant Panda National Park, and 21 studies of Northeast Tiger and Leopard National Park, for a total of 119 articles overall.

Finally, we reviewed the selected articles. For each article, we noted the conservation goals cited in the article as the ecological priorities of the relevant protected area, if any; the groups or communities identified as the main socioeconomic stakeholders of the protected area, if any; the tensions between these ecological priorities and socioeconomic stakeholders that were cited in the article; the synergies between these ecological priorities and socioeconomic stakeholders that were cited in the article; and the policy recommendations to resolve these tensions or take advantage of these synergies, as cited in the article. We then compiled lists of all ecological priorities, stakeholders, tensions, synergies, and policies cited by the literature for each park and identified the items that were cited the most frequently. In the sections below, we present our findings and explore the specific topics that are most frequently cited for each park, highlighting representative papers along the way in order to illustrate trends in the literature for each park.

## 4. Results

### 4.1. Sanjiangyuan National Park (SNP)

Out of 1139 articles returned in our initial literature search, we identified 28 articles that specifically relate to the interface between conservation efforts and socioeconomic interests in SNP (Table 1). Many of these studies (16 papers) focus on the park’s place in maintaining crucial ecosystem services in the Sanjiangyuan region [31,32], with several (7 papers) specifically centering on the park’s role in preserving populations of the endangered snow leopard [33,34].

**Table 1 ijerph-19-12778-t001:** Summary of major topics in the literature on Sanjiangyuan National Park.

Category	Specific Topic	Number of Papers	Citations
Ecological priorities	Preserve natural ecosystems	16	[18,23,31,32,35,36,37,38,39,40,41,42,43,44,45,46,47]
	Preserve snow leopard population	7	[33,34,37,48,49,50,51]
Stakeholders	Local communities	22	[18,23,31,33,34,35,36,37,38,39,40,42,45,48,49,50,51,52,53,54,55,56]
	Private businesses	5	[31,36,38,40,41]
Tensions	Economic stress from grazing restrictions	10	[18,23,31,35,39,45,48,49,52,56]
	Human–carnivore conflict	7	[23,37,48,49,50,52,53]
	Dissatisfaction with eco-migration/eco-compensation policies	4	[18,35,39,53]
	Unsustainable tourism practices	4	[36,38,41,46]
	Ecological damage from road construction	3	[34,36,46]
Synergies	Positive role of Buddhism	5	[23,37,43,49,50]
	Potential of ecotourism	5	[31,36,38,41,51]
Policy recommendations	Improve compensation schemes	9	[18,23,35,39,45,49,52,53,56]
	Invest in cultural tourism and ecotourism	6	[31,36,38,39,41,51]
	Provide residents with diversified income streams	6	[23,31,40,49,51,56]
	Improve environmental education and awareness	6	[23,35,38,39,50,56]
	Help residents protect food and livestock from carnivores	4	[23,49,52,53]

Our review of the literature reveals that local communities (22 papers) and private businesses (5 papers) are the most significant economic stakeholders in SNP. The socioeconomic needs of local residents, particularly Tibetan herders, frequently come into conflict with ecological initiatives within the park. For instance, the problem of human–carnivore conflict (HCC), which can take the form of livestock depredation or home break-ins by bears, is cited repeatedly in the literature (7 papers), with many residents specifically blaming ecological restoration programs for the high rates of property damage from bears in recent years [52]. Li et al. [49] found that higher incidences of HCCs negatively impact populations of snow leopards and raptors, which sometimes eat the poison traps locals have laid for bears and wolves. Researchers have also found that residents are dissatisfied with the government’s efforts to limit livestock grazing (10 papers) and with existing eco-migration and eco-compensation schemes (4 papers), which are perceived as harming the traditional lifestyles of herders [35]. Private businesses in the tourism sector also come into conflict with conservation efforts (4 papers). Buckley et al. [36] observed that some tourism companies have pushed for road construction in SNP, a controversial suggestion that is supported by some local authorities but decried by others as damaging to local ecosystems.

In contrast to these tensions, the literature also identifies Buddhist culture (5 papers) and the growth of the ecotourism sector (5 papers) as synergies between the park’s ecological and socioeconomic priorities. Buddhist monasteries in SNP have been shown to promote reverence of the natural environment and tolerance of predators among local residents, and they have played a key positive role in environmental education [37]. Sustainable tourism practices are similarly promising, as they can raise public awareness of conservation efforts while also creating opportunities for locals to earn non-pastoral income [38]. Buckley et al. [36] found that, relative to conventional tourists, ecotourists tend to value higher degrees of natural beauty in Sanjiangyuan. They also are less likely to demand high comfort levels that require a large degree of manipulation of the environment [36].

We identified four broad policy directions that researchers suggest can alleviate tensions between SNP’s ecological and socioeconomic interests. First, improved compensation schemes (9 papers) are shown to be effective in incentivizing herders to raise smaller flocks, thereby limiting landscape degradation [18,38]. Second, investing in education programs (6 papers), both general education and specifically environment-focused classes, can help locals learn about government conservation initiatives while giving them greater means to transition away from a pastoral lifestyle [35,39]. Third, investing in ecotourism (6 papers) allows the region to enjoy the economic benefits of tourism while avoiding the worst ecological damages of mass tourist infrastructure [31]. Some studies specifically explore the potential of an integration tourism or cultural tourism model, under which local residents enter the tourism industry by teaching visitors about traditional Tibetan practices and beliefs. This model can encourage locals to uphold cultural traditions while transitioning away from landscape-damaging traditional pastoral practices [31]. Finally, several researchers highlight the importance of reducing the frequency of human–carnivore conflict (4 papers), including through the use of government-funded livestock corrals [48] and bearproof steel bins for food [23].

### 4.2. Wuyishan National Park (WNP)

Out of the 811 articles returned by our initial literature search, we identified 12 empirical studies that explore the intersection between the development and conservation goals in WNP (Table 2). Most of the studies we reviewed (11 papers) identify the protection of regional biodiversity and local forests as the park’s primary ecological priority. Under WNP’s protection, the forests provide local residents and endemic species with a range of valuable ecosystem services, including climate regulation, biodiversity maintenance, and soil conservation [57].

**Table 2 ijerph-19-12778-t002:** Summary of major topics in the literature on Wuyishan National Park.

Category	Specific Topic	Number of Papers	Citations
Ecological priorities	Preserve forest ecosystems	11	[19,26,57,58,59,60,61,62,63,64,65]
Stakeholders	Local communities	12	[19,26,57,58,59,60,61,62,63,64,65,66]
	Tea industry	7	[19,26,57,58,59,60,62]
	Forestry industry	3	[58,60,63]
	Tourism industry	3	[26,59,61]
Tensions	Unsustainable tea cultivation practices	7	[19,58,59,62,65]
	Unsustainable tourism practices	4	[26,61,62,66]
	Lack of community participation in park	3	[63,64,66]
Synergies	Strong environmental consciousness of locals	3	[26,58,64]
	Economic benefits of conservation	1	[62]
Policy recommendations	Improve environmental education and awareness	8	[19,26,57,58,60,62,65,66]
	Invest in cultural tourism and ecotourism	3	[26,62,66]
	Increase community participation in park	3	[63,64,66]
	Strengthen returning tea to forest (RTTF) programs	2	[57,59]

WNP’s primary stakeholders are local residents and businesses, and their attitudes toward conservation and economic development in the region vary by industry. A significant portion of the literature focuses on tensions between conservation efforts and the tea industry (7 papers), as 80% of rural households in Wuyishan acquire the majority of their agricultural income from tea cultivation [58]. Some tea cultivation practices, such as pruning forest cover to maximize sunlight for tea bushes and replacing the soil of tea plantations with better soil from other areas, have come into conflict with new conservation policies [19]. The tourism industry (3 papers) is another source of tension in the region. Hsu et al. [66] documented evidence of overtourism at Jiuqu Stream, a popular scenic destination in WNP, where residents have expressed dissatisfaction with the management of trash, waste, and vehicle emissions. The researchers also found that the park struggles with shortages of cleaning and management personnel.

Our review, however, also identified promising synergies between the region’s socioeconomic needs and ecological goals. Research has shown that Wuyishan residents generally share a relatively strong environmental consciousness (3 papers). A survey conducted by He et al. [58] showed that many local residents near WNP recognize the importance of conserving key ecosystem services that relate to their own livelihoods. For instance, agricultural workers particularly value protecting sources of water, while those working in the tourism industry value air quality regulation. Future conservation efforts can build from these existing attitudes to increase their effectiveness and local support. In another encouraging sign of synergy between conservation and development, a study from Chang et al. [59] found that the ecosystem service value of WNP’s forests is greater than that of tea plantations or other types of vegetation land cover. This may indicate that restricting the tea industry in order to promote forest conservation (one of the goals of WNP) benefits the regional economy as a whole.

The policy recommendations most frequently cited in the literature are improved environmental education programs (8 papers), the development of the ecotourism and cultural tourism industries (3 papers), increased community participation in park management (3 papers), and returning tea to forest (RTTF) programs (2 papers). Environmental education targeted towards young local residents has been suggested as a policy solution to foster positive local attitudes towards conservation. In the long term, increasing the education of local children and their families about the importance of ecosystem regulation and biodiversity preservation almost certainly alleviates some of the tensions between the park and local communities [60]. Other researchers encourage investment in alternative forms of tourism, including ecotourism [61] and cultural tourism [62], that provide locals with new sources of income without contributing to the environmental damages of conventional tourism. The literature also suggests that giving Wuyishan residents a voice in the development and management of WNP may encourage them to take more initiative in pursuing conservation goals [63]. Finally, RTTF policies, in which tea farmers are subsidized to reforest certain plots of land, have been used in Wuyishan since 2008. Despite initial losses in tea profits from the policies, research shows that the long-term added ecosystem service value from restoring forests outweighs the costs. However, the RTTF policy remains unpopular among tea farmers because the ecological returns on investment are unclear and intangible. When examining alternative solutions, Chang et al. [59] suggested simply increasing subsidies for farmers and providing national funds for local governments to offset revenue loss.

### 4.3. Hainan Tropical Rainforest National Park (HTRNP)

Out of the 261 articles returned by our initial literature search, we identified 19 articles that focus on the interface between conservation efforts and local stakeholders in HTRNP (Table 3). These studies generally portray the park’s primary ecological priority as preserving Hainan’s rainforests (14 papers), specifically by conserving biodiversity [67] and protecting the source areas of the island’s major rivers [68]. The literature places particular emphasis on HTRNP’s role in protecting the remaining population of the endangered Hainan gibbons (5 papers). For instance, Deng et al. [69] emphasized the importance of Bawangling National Nature Reserve (now part of HTRNP) in maintaining old-growth forest for gibbon habitation. Qian et al. [70] highlighted the role of the Bawangling National Nature Reserve in educating locals about gibbon conservation.

**Table 3 ijerph-19-12778-t003:** Summary of major topics in the literature on Hainan Tropical Rainforest National Park.

Category	Specific Topic	Number of Papers	Citations
Ecological priorities	Preserve rainforest ecosystems	14	[4,20,67,68,71,72,73,74,75,76,77,78,79,80]
	Protect gibbon population	5	[69,70,73,81,82]
Stakeholders	Local communities	11	[20,67,69,70,72,75,76,77,78,81,82,83]
	Private businesses	9	[4,68,69,71,73,74,78,79,80]
Tensions	Ecological damage from large private industry (rubber, paper, real estate, etc.)	8	[4,68,71,73,74,78,79,80]
	Unsustainable tourism practices	8	[68,70,71,75,76,77,78,81,82]
	Ecological damage from individual households	5	[20,67,69,72,78,83]
Synergies	Positive attitude of locals toward conservation	4	[70,78,81,82]
Policy recommendations	Invest in reforestation programs	8	[67,68,69,71,73,74,79,80]
	Impose stricter logging restrictions	6	[4,69,71,74,79,80]
	Improve environmental education and awareness	5	[70,75,77,81,82]
	Require land use reform	4	[4,72,78,80,83]
	Invest in ecotourism	3	[76,77,78]

The main economic stakeholders identified in the literature on HTRNP are local residents (11 papers) and private businesses (9 papers). Economic activities by both groups have been identified as threatening HTRNP’s ecology. The literature shows that locals living near HTRNP tend to have low income levels [4,83] and to rely on land-intensive economic activity for their livelihoods, including the cultivation of rubber plantations, crop farming, and livestock rearing [83]. Researchers have documented ecological damage resulting from unsustainable practices by locals (5 papers), including planting monoculture crops [20], and cutting into forest land to expand the land area occupied by private homes and associated economic activities [67]. Rainforest land has also been logged to create space for farmland and rubber plantations, the latter of which tends to damage critical ecosystem services such as soil retention and flood prevention [4]. In addition to economic activity by local households, private industry on Hainan, a Special Economic Zone that attracts large amounts of domestic and foreign investment, is a key source of tension between HTRNP’s ecological and socioeconomic priorities. The burgeoning tourism industry (8 papers) has contributed to an expanding need for real estate land, which in turn drives deforestation of the island’s rainforests [82]. Infrastructure development projects in Hainan, such as the paving of highways, have also led to environmental degradation [69]. 

The literature, however, also identifies synergies between HTRNP’s ecological and economic needs. In particular, the broadly positive outlook of residents on environmental conservation (4 papers) indicates a strong potential for collaboration between park authorities and local communities. In a survey conducted in 26 towns near HTRNP, Ma et al. [81] demonstrated that public awareness campaigns have successfully raised local consciousness about gibbon conservation. According to their findings, residents would be receptive to further outreach and education. W. Han et al. [82] also found that HTRNP residents that identify with traditional Chinese cultural values also tend to feel empowered to enforce environmentally conscientious behavior among tourists.

Researchers have proposed a range of policies to align the region’s socioeconomic needs with conservation goals. First, many studies highlight the importance of reforestation programs (8 papers). For instance, Zhao et al. [20] framed subsidized tree planting programs as a sustainable way to supplement the incomes of local residents. Second, the literature shows that land use reform (4 papers) can curb monoculture farming and promote key ecosystem services, benefiting both local livelihoods and conservation efforts [4,83]. Finally, empowering local communities and using them as a way to spread environmental awareness (5 papers) can alleviate the burden on park rangers by placing greater obligations on locals themselves to protect the land [70,82]. These mass awareness campaigns can simultaneously improve the behavior of locals and tourists [81].

### 4.4. Giant Panda National Park (GPNP)

Out of the 1789 articles returned by our literature search, we identified 39 studies that deal with interactions between local stakeholders and conservation programs in GPNP (Table 4). Our review found that the literature most often highlights the park’s role in promoting giant panda habitat preservation and habitat connectivity (33 papers). Huang et al. [84] specifically emphasized that GPNP brings a large share of the panda’s range, which has long been administered by a patchwork of separate reserves, under a single authority. Many studies (13 papers) also highlight the park’s role in protecting other vulnerable species within its boundaries, such as the red panda and the golden snub-nosed monkey [85,86,87].

**Table 4 ijerph-19-12778-t004:** Summary of major topics in the literature on Giant Panda National Park.

Category	Specific Topic	Number of Papers	Citations
Ecological priorities	Protect giant panda populations	33	[21,84,85,86,87,88,89,90,91,92,93,94,95,96,97,98,99,100,101,102,103,104,105,106,107,108,109,110,111,112,113,114,115]
	Protect other vulnerable species	13	[85,87,88,89,90,92,93,94,98,111,116,117,118]
	Prevent land degradation	11	[86,95,99,104,105,106,107,109,112,119,120]
Stakeholders	Local communities	24	[21,84,85,86,87,88,89,90,91,93,94,95,96,102,103,106,107,108,109,110,112,113,116,119,121]
	Private businesses	13	[91,92,95,96,98,99,102,103,104,106,113,119,120]
Tensions	Ecological damage from logging, ranching, and agriculture	24	[21,84,86,87,91,93,94,95,96,97,98,99,103,104,106,107,108,109,110,112,113,114,115,119,120]
	High cost of conservation for locals	8	[21,85,89,90,96,97,116,121]
	Pollution (air, waste, and heavy metals)	6	[91,92,93,100,101,117]
	Unsustainable tourism practices	4	[92,95,113,116]
	Loss of tradition	2	[88,116]
Synergies	Potential for ecotourism development	8	[85,88,89,90,94,95,116,121]
	Willingness of locals to pay for conservation	5	[87,88,93,94,116]
Policy recommendations	Restrict grazing, planting, and construction in/around park	19	[86,91,92,95,98,100,101,102,103,104,106,107,108,109,112,115,117,119,120]
	Invest in ecotourism and increase tourism regulation	9	[84,85,88,89,90,92,95,96,113,116]
	Improve ecological compensation and migration schemes	8	[84,85,88,89,90,94,96,97,121]
	Streamline local input in park management	6	[85,88,93,94,116,121]
	Expand park boundaries	3	[96,114,115]

Our review of the literature reveals that rural households (24 papers) and private businesses (13 papers) have economic interests in GPNP that frequently come into conflict with conservation goals in several ways. First, both small-scale and large-scale agriculture and ranching activities in the vicinity of the park (24 papers) have been shown to contribute to the disturbance of panda populations. In particular, the conversion of collective forest into plantations by local communities, which is currently permitted in GPNP, threatens to harm panda habitat connectivity if it is not restricted [102]. Likewise, the practice among local communities of allowing livestock to graze in panda habitat zones also has been shown to cause significant panda behavioral disturbance [85,99,109]. Second, the relatively high costs associated with ecological conservation for individual households (8 papers) have created tension within local communities inside GPNP. A survey of residents living in giant panda reserves from Ma et al. [89] found that the costs of living in protected areas, including damages caused by wildlife and restricted access to natural resources, outweigh the compensation that residents receive from the government for these costs. In analyzing these issues, Shi et al. [116] showed that restrictions imposed by the park increase household competition for resources and often negatively affect social trust in these local communities. Finally, both households and businesses participate in unsustainable practices related to the tourism industry (4 papers). Y. Zhao et al. [92] documented a range of threats to panda populations that are correlated with the expanding tourism industry in the park, including increased air pollution and road construction. On a smaller scale, Shi et al. [116] found that, in response to a tourist-driven rise in demand for medicinal materials and wild vegetables, local residents increasingly travel into panda habitats to collect herbs, causing habitat destruction.

Despite these tensions, the literature reveals promising foundations for collaboration between locals and park authorities. Research shows that promoting ecotourism (8 papers) can increase the incomes and professional skills of local residents while mitigating damage from tourism on the environment [88]. Ma [90] also found that ecotourism increases the willingness of communities to participate in conservation by 154%. Furthermore, researchers have found that residents of GPNP tend to be willing to pay for conservation programs (4 papers). In a survey by Zhang et al. [93], rural households near GPNP reported a high willingness to pay for types of conservation that are related to their own livelihoods. For example, households that own tracts of forestland tend to be more willing to pay for forest vegetation restoration. These positive attitudes toward conservation may represent opportunities for park authorities in their future interactions with local communities.

The major policy recommendations that we identified in the literature include curtailing grazing and other economic activity around GPNP (19 papers), increasing investment in and regulation of the ecotourism industry (9 papers), and improving ecological compensation schemes for local residents (8 papers). Restricting economic activity that is correlated with panda behavioral disturbance is an obvious priority for panda conservation, and strictly limiting the construction of new roads [91] is an important first step toward this goal. However, as Wang et al. [86] pointed out, directly suppressing harmful grazing and planting behavior through strictly enforced rules likely alienates residents and increases conservation costs. Therefore, it is preferable to indirectly encourage residents to move away from traditional livelihoods by promoting more eco-friendly industries, such as ecotourism. In terms of mitigating the effects of tourists on the local environment, key regulations proposed by researchers include limiting road construction inside the park [91], raising fines for littering, increasing patrols of the park, and providing clear signage to prevent tourist incursion into vulnerable areas [116]. Finally, to further reduce the tension between wildlife and households, Ma [90] recommended improving existing ecological compensation structures to incentivize communities to maintain ecological public welfare forests. These compensation programs currently have some public support but require more consistency and oversight. Ma’s research also showed that residents would be willing to accept lower compensation levels if more ecological or ecotourism jobs became available, highlighting the potential for these policies to work in tandem.

### 4.5. Northeast Tiger and Leopard National Park (NTLNP)

Out of 1334 articles returned by initial literature search, we identified and reviewed 21 studies related to interactions between human stakeholders and ecological priorities in NTLNP (Table 5). Many of the papers we reviewed (12 papers) focus on NTLNP’s role in preserving China’s Amur tiger and Amur leopard populations. The park serves to prevent the direct killing of tigers and leopards, whether from poaching or retaliatory killing [122,123]. The NTLNP also seeks to limit human disturbance in wetland and forest habitats that are important to the Park’s carnivore and prey population growth [124,125].

**Table 5 ijerph-19-12778-t005:** Summary of major topics in the literature on Northeast Tiger and Leopard National Park.

Category	Specific Topic	Number of Papers	Citations
Ecological priorities	Protect Amur tiger and Amur leopard populations	12	[122,123,125,126,127,128,129,130,131,132,133,134,135]
	Protect other vulnerable species	8	[122,126,127,128,129,131,133,134]
	Prevent wetland degradation	3	[124,136,137]
Stakeholders	Local communities	12	[122,124,125,126,127,128,129,130,131,135,136,138,139]
	Private businesses	4	[126,127,130,136]
Tensions	Ecological damage from agriculture, grazing, and related sources	15	[124,125,126,127,128,129,130,131,132,134,136,138,140,141,142]
	Economic stress from conservation policies	4	[122,127,128,130]
Synergies	Availability of labor following state forest reform	1	[130]
Policy recommendations	Restrict livestock grazing in park	10	[122,125,127,128,129,131,132,134,138,142]
	Strengthen enforcement of park rules	9	[122,124,126,127,128,130,131,136,142]
	Increase ecological compensation schemes	9	[122,124,125,128,129,130,132,136,141]
	Expand park boundaries	5	[122,127,128,129,133,135]

The primary economic stakeholders around NTLNP are local residents (12 papers), many of whom rely on land-intensive livestock rearing for their livelihoods [124,128]. Livestock and agriculture (15 papers) are frequently cited in the literature as major sources of behavioral disturbance among tigers and their prey in the Changbai Mountains [125,141,142]. Ecological damage from agriculture also has been documented in the Tumen River Basin, another region encompassed by the national park. Liu et al. [124] reported that nearly half of the area’s wetland loss in recent decades is a result of agricultural encroachment. Zhang et al. [93] found that the conversion of marshes, rivers, and paddy fields into farmland has led to a significant loss in carbon sequestration and water yield. In addition to agriculture, the logging industry has historically played an important role in the vicinity of the park. Recent restrictions on logging imposed by the government have put significant economic stress on residents that once relied on the sector for income [130].

Unlike the other four national parks, the literature around NTLNP notably does not highlight areas of potential synergy between socioeconomic interests and conservation goals. In particular, the promise of ecotourism, which arises repeatedly in studies about the other four parks, appears to have been largely ignored so far in the literature about NTLNP, perhaps because of the park’s remoteness. However, Sun and Geng [130] noted that the economic vacuum left in the wake of the government’s logging restrictions represents an opportunity to retrain loggers for work in industries that promote conservation, thereby meeting the region’s ecological and economic needs simultaneously.

Despite the lack of promising synergies in the literature, we identified three broad policy recommendations that arise frequently in the literature. First, research shows that there is a need to ban or restrict livestock grazing within park boundaries (10 papers) in order to mitigate the disturbance of the carnivore populations and to minimize livestock depredation [138]. Second, the literature recommends increasing the enforcement of park rules (9 papers) to protect the biodiversity within the park, including by retraining forest rangers to participate in biodiversity preservation and monitoring [122]. Third, several studies present expanding the park’s current boundaries (5 papers) as a crucial step to sustain healthy tiger and leopard populations [127,133,135].

## 5. Discussion

While China’s new national park system is a promising move toward protecting natural resources and ecosystems on a national scale, significant steps can be taken in each park to promote sustainable development, as our review shows. To meet the United Nations Sustainable Development Goals (SDGs) as laid out in the 2030 Agenda for Sustainable Development [3], it is crucial for China to design park policy with each park’s unique characteristics in mind. The tensions and synergies between conservation efforts and socioeconomic stakeholders take different forms in each park. Individualized, data-based solutions would be most effective when addressing the specific issues that involve the different parks. Moreover, while the specific synergies and policy recommendations identified in the literature can provide a blueprint to balance conservation and development in each park, evidence from other PA systems that have attempted to strike this balance in other parts of the world shows that even effective management strategies only have fleeting positive effects for development unless implemented effectively. Specifically, strong community involvement in planning in management, empowered park authorities, and long-term commitment to change are necessary to bring these park-specific policies to fruition [15,16,17].

At the same time, there remain broad measures that China can take on a larger scale in order to bring the national park system as a whole more into accord with the SDGs. Doing so promotes the socioeconomic development of human communities in and near the park as a goal parallel to and inseparable from ecological conservation, thereby forming a robust basis for ecological solidarity [15]. Specifically, our review reveals four common themes that apply to all five of China’s first national parks. These themes can guide park policy moving forward and point to promising directions for future research.

First, research from all five parks highlights the importance of striking a balance between traditional ways of life and ecological preservation. Broadly speaking, the national park system is designed to align with SDG 15, which requires countries to “protect, restore and promote sustainable use of terrestrial ecosystems, sustainably manage forests … and halt and reverse land degradation and halt biodiversity loss.” However, all five parks would benefit from closer attention to SDG 15.9: “Integrate ecosystem and biodiversity values into national and local planning, development processes, poverty reduction strategies and accounts” [3]. To this end, where possible, local governments can improve the efficiency or regulation of traditional industries so that they can coexist with sustainability initiatives. Examples include providing herders with protective livestock corrals to reduce depredation from carnivores in SNP [48] and encouraging medicinal herb growing rather than wild herb collection in GPNP [116]. When traditional industries need to be scaled back in order to preserve each park’s ecology and to conform to conservation guidelines, studies from all five parks show that it is important to offer local residents alternative, sustainable income streams, such as participating in the ecotourism industry [38,62,88] or serving as a park ranger [20,51,81,82,94,130]. Ecological compensation policies, which can offset lost profits and reduced incomes associated with conservation initiatives, have already been shown to be effective in SNP [45], WNP [59], and GPNP [90], but further research is needed to determine if they can be useful in the other parks (e.g., HTRNP and NTLNP). 

Second, infrastructure development needs to be aligned with the local ecology in all five parks, particularly given the higher visitor numbers that are likely to accompany each location’s new national park status. This theme echoes SDG 9, which requires countries to “build resilient infrastructure [and] promote inclusive and sustainable industrialization” [3]. For example, road construction in parks needs to be carefully weighed against ecological factors and prohibited where necessary. This is particularly important in parks where research has shown road construction to result in environmental pollution and habitat fragmentation, namely SNP [36], HTRNP [69], GPNP [91], and NTLNP [141]. Further research should be conducted to determine whether these risks exist in WNP as well.

Third, steps need to be taken to promote education and environmental awareness among park residents. Doing so would bring China’s national park system further into alignment with SDG 12.8: “By 2030, ensure that people everywhere have the relevant information and awareness for sustainable development and lifestyles in harmony with nature” [3]. To this end, the importance of ecological public awareness campaigns [38,60,81], as well as job training programs [61,122,130], is well-established in the literature. However, some researchers also suggest that promoting general education and school attendance in communities within and around the national parks can lead to positive environmental outcomes in SNP [39]. The preliminary nature of this important topic means that more research should be conducted to determine whether these policies would be successful in the other parks as well, and how such policies should be best implemented.

Finally, evidence from the majority of the parks shows that supporting the development of a sustainable tourism and/or cultural tourism industry within the parks can promote economic development and ecological conservation simultaneously. In addition to these benefits, promoting these industries would help satisfy the requirement in SDG 12b that countries “Develop and implement tools to monitor sustainable development impacts for sustainable tourism that creates jobs and promotes local culture and products” [3]. Investment in ecotourism can be paired with caps on daily visitors and improved enforcement of park rules to limit the ecological damages when the number of visitors begin to rise [66,94,122]. Cultural tourism in SNP and WNP has also been shown to be an effective way to promote cultural traditions while transitioning local communities away from unsustainable (often traditional) livelihoods [31,38,62]. NTLNP is the only national park where the potential for ecotourism has not yet been assessed. This represents an obvious opportunity for future research.

## 6. Conclusions

China’s ambitious new system of national parks has the potential to vastly improve the country’s ability to manage and conserve its environmental resources. However, the expansion of such high profile public parks also causes new challenges in the relationship between park authorities and local communities. To resolve these challenges, China needs to closely monitor findings and warnings from empirical research conducted within the parks and adjust its policies accordingly.

Our review of the existing literature shows that several key tensions and synergies between conservation goals and socioeconomic stakeholders exist widely in the new park system. First, the traditional livelihoods of many park residents contradict conservation rules in all five parks, a problem that can be addressed with increased investment in park-specific regulation and ecological compensation programs in the future. Second, infrastructure development is difficult to reconcile with conservation priorities, and there is evidence that limitations on road construction will ultimately be necessary in most or all the national parks. Third, environmental education and public awareness campaigns have been shown to be beneficial for both conservation and development in all five parks, although further research is needed to determine whether investments in general education can also have a similar effect. Finally, while the environmental damage associated with conventional tourism is well documented in most of the parks, our review shows that promoting the development of ecotourism or cultural tourism industries can have benefits for both conservation and development in all five parks.

By synthesizing past findings about tensions and synergies between park authorities and local stakeholders in China’s national parks, this narrative review contributes insights that can guide future policymaking and research. Park authorities need to respond to the park-specific challenges and opportunities identified in our review (Section 4.1, Section 4.2, Section 4.3, Section 4.4, Section 4.5) in order to ensure that conservation and development continue to be promoted in the existing five parks. At the same time, they need to learn from the trends identified in the existing parks (Section 5) as new national parks continue to be added to the system in the coming years. Likewise, researchers operating at the intersection between protected areas and local communities need to both monitor how park-specific issues evolve over time and to fill in the gaps in the literature that we identified (Section 5), including the lack of data on the effects of general education on local participation in conservation and the need for research into road construction and ecotourism development in several of the parks.

However, we acknowledge two limitations in this review: First, this article only considers research that was originally published in English, despite the large body of important literature about national parks and local communities that exists in Chinese. Future studies should take Chinese-language research into account in order to gain a more complete understanding of the existing literature. Second, while the studies reviewed for this article are limited to a ten-year publication window, the rapid pace of changes in China’s ecology and protected area management system means that new developments will soon arise that are not reflected in this paper. It will be necessary for future researchers to continue to monitor the actual situation in the parks and the changes that occur as China’s national park system matures.

## Figures and Tables

**Figure 1 ijerph-19-12778-f001:**
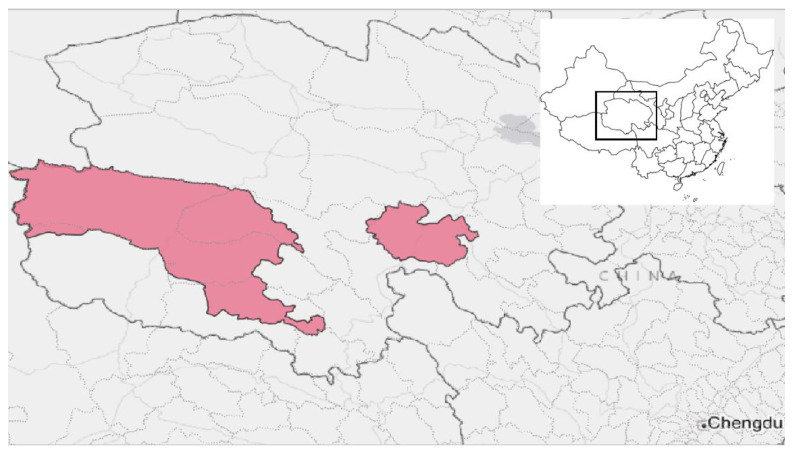
Location of Sanjiangyuan National Park in Qinghai Province, China. All figures use public-domain base maps from Stamen and Wikimedia Commons and location data from the National Forestry and Grasslands Administration.

**Figure 2 ijerph-19-12778-f002:**
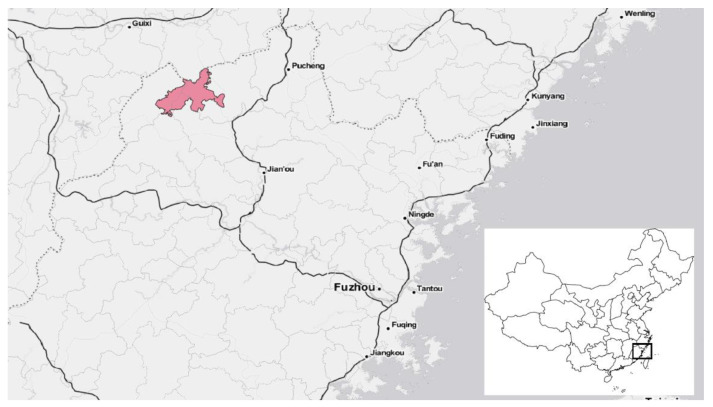
Location of Wuyishan National Park in Fujian Province, China.

**Figure 3 ijerph-19-12778-f003:**
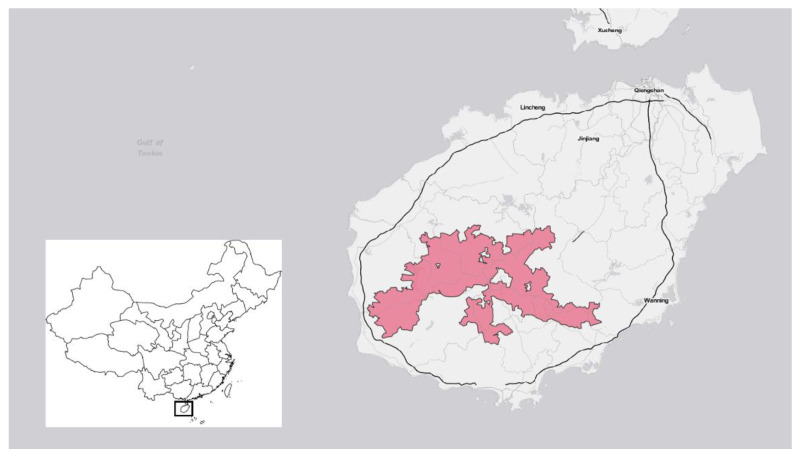
Location of Hainan Tropical Rainforest National Park in Hainan Province, China.

**Figure 4 ijerph-19-12778-f004:**
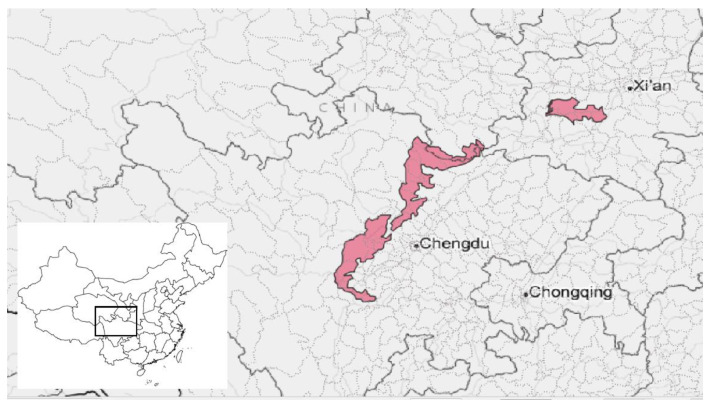
Location of Giant Panda National Park in Sichuan, Gansu, and Shaanxi Provinces, China.

**Figure 5 ijerph-19-12778-f005:**
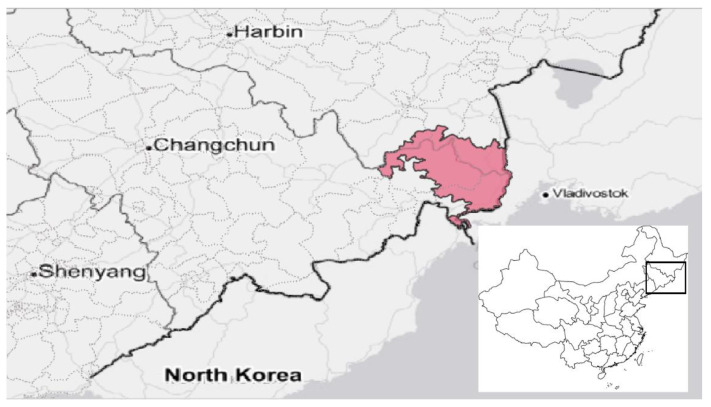
Location of Northeast Tiger and Leopard National Park in Jilin and Heilongjiang Provinces, China.

## Data Availability

Not applicable.

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
