# Peer review of "Challenges and Opportunities in Aligning Conservation with Development in China’s National Parks: A Narrative Literature Review"

_ijerph, 2022, doi:10.3390/ijerph191912778_

Round 1
Reviewer 1 Report
I thank the authors for giving me the opportunity to review their work dealing with protected areas in China, from a social-ecological systems perspective.
The article reports an interesting desk review of the literature about the five parks in China, however some changes are necessary in order to better understand the method applied for the review.
Section 2 needs to be revised: an introductory paragraph on protected areas in China is needed, providing more information on the system, national laws governing them, if there are specific national or local authorities for management, etc. Subsequently, the authors can talk about the five protected areas, possibly specifying for each of them the IUCN category to which they belong, managing authority, extension, year of establishment, etc. The image should be placed after the text and not the other way around (as it is now).
In the Results section, however, it is not clear how the categories and specific topics were selected, it is necessary to explain in materials and methods how the table was defined.
The discussions are quite interesting because they outline guidelines for managing and defining effective policies for protected areas. However, the recommendations aimed at achieving the objectives of the 2030 Agenda could be further explored.
I advised to review the article with "major revision" but it is actually halfway between this evaluation and "minor revision"
Reviewer 2 Report
Paper review
The paper aims to assess challenges and opportunities in aligning conservation with development in China's national parks through a literature review. Although the paper's objectives are well set, there is no broader/conceptual discussion on the conservation x development issues worldwide and in China. A literature review of those topics is required to guide the methodology, results and discussion.
It is suggested to point out a research gap and the paper's contributions to the field of conservation.
The results and discussion presented report what is in previous papers and relate this information without critically addressing those results against the existing conservation x development issues literature.
The conclusion should be reviewed, highlighting your findings and main discussion points, contributions, limitations, and the scientific value of your paper.
Round 2
Reviewer 1 Report
I want to thank the authors who, in some paragraphs, extensively revised the paper. In my opinion the final version of the manuscript acquired more scientific consistency.
Please check one more time the English, some sentences or words need a review
Reviewer 2 Report
The authors have addressed all topics raised in the review process and present a consistent paper. Congratulations on the effort and result!